# Optimizing the Ultrasound Image Quality of Carotid Artery Stenosis Patients via Taguchi's Dynamic Analysis and an Indigenous Water Phantom

Kai-Yu Hsiao [1,2,3], Chih-Sheng Lin [4], Wan-Ming Li [5], Shih-Hsun Huang [6,7], Yu-Ting Cho [6], Bing-Ru Peng [6], Lung-Kwang Pan [6] and Lung-Fa Pan [6,8,*]

1    Department of Surgery, Taichung Armed-Forces General Hospital, Taichung 411, Taiwan
2    Division of Thoracic Surgery, Department of Surgery, Tri-Service General Hospital, National Defense Medical Center, Taipei 114, Taiwan
3    Graduate Institute of Medical Science, National Defense Medical Center, Taipei 114, Taiwan
4    Department of Radiology, BenQ Medical Center, The Affiliated BenQ Hospital of the Nanjing Medical University, Nanjing 211166, China
5    Department of Biomedical Imaging and Radiological Science, China Medical University, Taichung 404, Taiwan
6    Department of Medical Imaging and Radiological Science, Central Taiwan University of Science and Technology, Taichung 40601, Taiwan
7    Department of Nursing, Taichung Armed Forces General Hospital, Taichung 411, Taiwan
8    Department of Cardiology, Taichung Armed Forces General Hospital, Taichung 411, Taiwan
*    Correspondence: pan5302@yahoo.com.tw

**Abstract:** This study optimized the ultrasound image of carotid artery stenosis using Taguchi dynamic analysis and an indigenous water phantom. Eighteen combinations of seven essential factors of the ultrasound scan facility were organized according to Taguchi's $L_{18}$ orthogonal array. The seven factors were assigned as follows: (1) angle of probe; (2) signal gain; (3) resolution vs. speed; (4) dynamic range; (5) XRES; (6) zoom; (7) time gain compensation. An indigenous water phantom was customized to satisfy the quantified need in Taguchi's analysis. Unlike the conventional dynamic Taguchi analysis, an innovative quantified index, the figure of merit (FOM), was proposed to integrate four specific quality characteristics, namely (i) average difference between the practical scan and theoretically preset area (78.5, 50.2 and 12.6 mm$^2$) of stenosis, (ii) standard deviation of the average, (iii) practical scan's sensitivity β to various stenosis diameters (10, 8, and 4 mm), and (iv) correlation coefficient $r^2$ of the linear regressed sensitivity curve. The highest value (*FOM* = 0.413) was furnished by the optimal combination of factors on 18 groups under study, yielding high $r^2$ and low β or standard deviation values and the best quality of ultrasound images for the further clinical diagnosis. The comparison between FOM and the conventional signal-to-noise (S/N) ratio in Taguchi's analysis revealed that FOM compiled more quality characteristics that were superior by nature to fulfill the practical need in clinical diagnosis. The alternative choice in ultrasound scan optimization can be based on stenosis diameter variation from a different perspective to be explored in the follow-up study.

**Keywords:** carotid artery stenosis; ultrasound image quality; Taguchi's dynamic analysis; orthogonal array; figure of merit (FOM); signal-to-noise (S/N) ratio; water phantom

## 1. Introduction

Carotid ultrasound scans can be split into the following two main categories: grayscale B-mode and compound color Doppler mode. The former mode is mainly used to survey the thickness of the intima–media thickness (IMT), the size and location of atherosclerotic plaques, the nature of echoes, and the surface of the blood vessels in the neck [1]. The latter is applied to survey blood flow velocity in vessels according to the Doppler effect. Involving color or pulse waves [2], it is positioned as the main noninvasive imaging modality used

to monitor patients after endarterectomy or stenting [3,4]. Meanwhile, the grayscale B-mode is widely used in the routine ultrasound examination of carotid arteries to detect abnormalities and flow-limiting artery stenosis for revealing cranial and external carotid artery disease in clinical diagnosis.

It is essential to optimize the ultrasound scan images in routine diagnosis since its noninvasive nature always prioritizes for medical staff to catch the general idea of stenosis syndrome. Several researchers have noted the importance of acquiring high-quality ultrasound scanned images and proposed solutions via various approaches. Lal et al. [5] adopted computer-aided analysis, while Santos [6] and Araki et al. [7] proposed new algorithms to improve the resolution of scanned images by their postprocessing. Alternatively, Yau et al. [8] developed a tissue-like phantom for imaging optimization from a practical survey approach, while Ponnle et al. [9] applied a customized silicone-rubber tube phantom to optimize the carotid arterial wall image. Poree et al. [10] combined polyvinyl alcohol cryogel phantoms with optimization of ultrasound scans based on carotid plaque geometries in MRI images. Finally, Kingstone et al. [11] created a simplified five-tissue-like phantom, effectively simulating various diseased plaque segments. However, optimizing image quality of grayscale B-mode carotid ultrasound scans of carotid artery stenosis patients remains a challenging problem, for which a solution would significantly enhance the diagnostic accuracy. A narrow profile of carotid artery stenosis (in contrast to the cardiac one) is often ignored by radiologists. Based on previous findings of our research group [12–15], this study attempts to solve it by using an indigenous water phantom and dynamic Taguchi analysis based on the figure of merit (FOM) approach, in contrast to the earlier applied static one based on signal-to-noise (S/N) ratio [12].

In this study, a silicon tube was customized to imply the blood vessel, and an indigenous water phantom was set up to create a complete hydraulic circulation system simulating human carotid artery stenosis syndrome. The quantitative evaluation of the scanned image quality was accomplished by Taguchi's dynamic analysis with an innovative definition of the figure of merit (FOM). The Taguchi $L_{18}$ orthogonal array adopted in this study was well-reputed for its rapid and effective feature in collecting practical data. Accordingly, seven factors were assigned in two or three levels to organize the unique eighteen combinations for practical data collection. Unlike static Taguchi's analysis, the ANOVA (statistic of variance) was not adopted to evaluate the derived outcomes in the dynamic one. However, the follow-up verification and correlated elaborations in both the experiment and data analysis were included in the discussion.

## 2. Materials and Methods

### 2.1. Study Design

The TAFGH Institutional Review Board committee approved the study with credential No. TSGHIRB 2-105-05-089, and the requirement for informed consent was waived.

### 2.2. Dynamic Taguchi's Analysis

High-quality characteristic systems are effectively optimized via dynamic Taguchi's analysis, which arranges special orthogonal arrays under various boundary conditions and assess contributions of each significant factor by performing a limited number of tests. It also envisages tuning-off the optimal combination of factors controlling the scanned image quality from insignificant ones and the environment. The figure of merit (FOM) was comprehensively compiled to ensure the factor settings for the optimal ultrasound scan imaging protocol [15,16]. A flow chart can further imply the process, as illustrated in Figure 1. As depicted, the project starts by presetting the demanded purpose, then ensuring the number of factors. It is recommended to have multiple measurements in each group to increase the number of degrees of freedom (DOF) for the follow-up survey. Since Taguchi optimization is also reputed for its solid suggestions for solving practical problems, the preliminary recommendation of Taguchi's analysis needs to be further verified and testified through clinical confirmation to ensure its accuracy. If it fails to pass the clinical verification,

this recommendation needs to be retrieved by considering the cross-interactions among factors to eliminate their interference and propose the refined recommendation again to satisfy the clinical demand exactly.

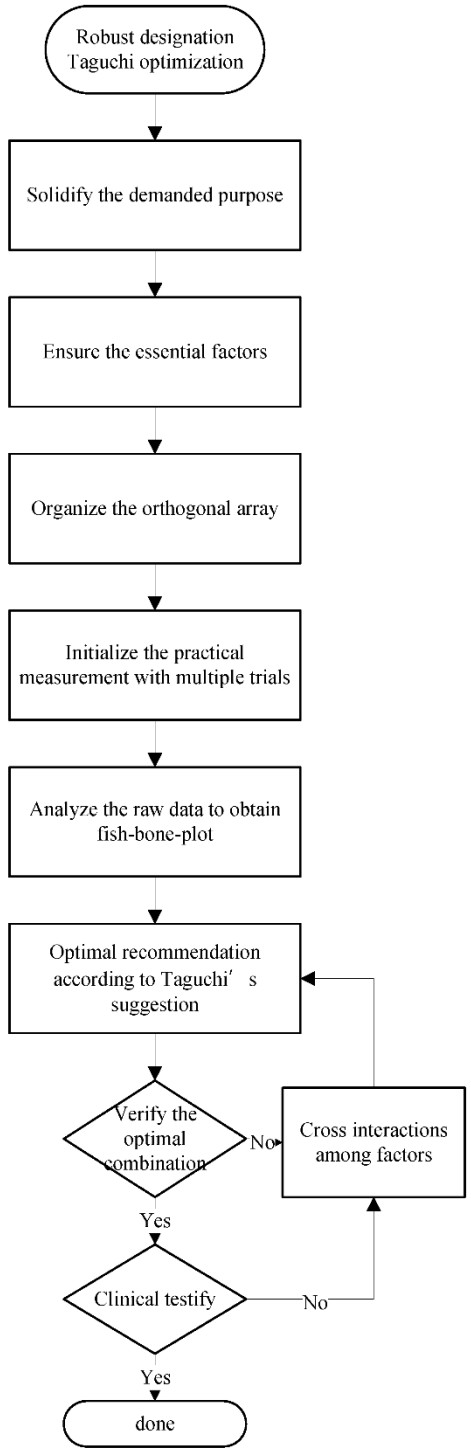

**Figure 1.** The flowchart of workload illustrates how researchers apply Taguchi's optimization methodology to ultrasound facilities in practice.

### 2.3. Orthogonal Arrays

The following seven factors were assigned in ultrasound scan protocol settings for carotid artery stenosis: (1) angle of probe, (2) signal gain, (3) resolution vs. speed, (4) dynamic range, (5) XRES (Philips XRES, a diagnostic ultrasound image-processing technique

enhancing subtle diagnostic data by helping the human eye to better perceive patterns, i.e., an adaptive real-time algorithm provided by Philips), (6) zoom, and (7) time gain compensation. Thus, a complete set of 1458 ($2^1 \times 3^6$) combinations of seven factors was analyzed (each factor was categorized into two or three possible levels). The arrangement of samples into only eighteen groups via Taguchi's analysis ensured the same confidence level of results as the conventional optimization of processes [17]. Table 1 illustrates a typical Taguchi $L_{18}$ ($2^1 \times 3^6$) orthogonal array, with digits in columns indicating practical arrangements (levels) of factors A–G. Seven factors (each bearing two or three levels) of Philips Affiniti 70 ultrasound scan facility with a L12-3 probe as used in the proposed ultrasound scan protocol (freq. 3–12 MHz), according to Taguchi's settings in Table 1, are summarized in Table 2: (A) angle of probe—in practice, it is adjusted so as to ensure the best echo reflection, despite the theoretical guidance of vertical or nearly vertical probe positioning to suppress the uncertainty; (B) signal amplification gain—after penetrating the tissue, the ultrasonic signal returns to the transducer and is subjected to uniform amplification (signal gain). Instead of increasing the image brightness, a uniform margin should whiten the image on the screen for easier detection of details; (C) resolution vs. speed—the dynamic recognizing system moderates either resolution or scanning speed by consequently optimizing the resolution and speed (R1 mode), or vice versa (S1 mode), or compromising them simultaneously (RS mode). In routine diagnosis, the system is preset at RS mode by default; (D) dynamic range, the preset range of B-mode grayscale ultrasound scan images; (E) XRES, which is a diagnostic ultrasound image-processing technique enhancing subtle diagnostic data by helping the human eye to better perceive patterns [18]. In addition, the original preset is scenario 2; (F) zoom, a function used to observe a specific area in more detail. It enlarges the corresponding area image using simple algorithms, such as linear interpolation; (G) time gain compensation, a setting applied in diagnostic ultrasound imaging to account for tissue attenuation. It increases the received signal intensity with depth, reducing the artifacts in the uniformity of the B-mode image intensity. Moreover, all other presets were factory default (frequency of 3.6 MHz, band width of 40–80 Hz, and imaging method rate of 66 mm/s).

**Table 1.** The standard Taguchi's $L_{18}$ ($2^1 \times 3^6$) orthogonal array, where the numbers in each column, except the first one, indicate the specific factor (A–G) level or practical arrangement.

| Group | Factor | | | | | | |
|---|---|---|---|---|---|---|---|
| | **A** | **B** | **C** | **D** | **E** | **F** | **G** |
| 1 | 1 | 1 | 1 | 1 | 1 | 1 | 1 |
| 2 | 1 | 1 | 2 | 2 | 2 | 2 | 2 |
| 3 | 1 | 1 | 3 | 3 | 3 | 3 | 3 |
| 4 | 1 | 2 | 1 | 1 | 2 | 2 | 3 |
| 5 | 1 | 2 | 2 | 2 | 3 | 3 | 1 |
| 6 | 1 | 2 | 3 | 3 | 1 | 1 | 2 |
| 7 | 1 | 3 | 1 | 2 | 1 | 3 | 2 |
| 8 | 1 | 3 | 2 | 3 | 2 | 1 | 3 |
| 9 | 1 | 3 | 3 | 1 | 3 | 2 | 1 |
| 10 | 2 | 1 | 1 | 3 | 3 | 2 | 2 |
| 11 | 2 | 1 | 2 | 1 | 1 | 3 | 3 |
| 12 | 2 | 1 | 3 | 2 | 2 | 1 | 1 |
| 13 | 2 | 2 | 1 | 2 | 3 | 1 | 3 |
| 14 | 2 | 2 | 2 | 3 | 1 | 2 | 1 |
| 15 | 2 | 2 | 3 | 1 | 2 | 3 | 2 |
| 16 | 2 | 3 | 1 | 3 | 2 | 3 | 1 |
| 17 | 2 | 3 | 2 | 1 | 3 | 1 | 2 |
| 18 | 2 | 3 | 3 | 2 | 1 | 2 | 3 |

**Table 2.** Seven factors (each bearing two or three levels) used in the proposed ultrasound scan protocol, according to Taguchi's settings in Table 1.

| Factor | Levels | | |
|---|---|---|---|
| | **1** | **2** | **3** |
| (A) angle of probe (degree) | 30 | 0 | |
| (B) signal amplification gain (dB/cm) | 50 | 55 | 60 |
| (C) resolution vs. speed | R1 | RS | S1 |
| (D) dynamic range (dB) | 40 | 50 | 60 |
| (E) XRES (scenario) | 1 | 2 | 3 |
| (F) zoom | 0.8 | 1.0 | 1.2 |
| (G) time gain compensation | −1 | 0 | +1 |

*2.4. The Customized Water Phantom*

Our research group designed several water phantoms customized to satisfy multiple factor settings in the Taguchi's analysis for various medical applications. These included line pair gauges [13,19], a V-shaped line gauge [14,15], and the latest water phantom [12] composed of two main compartments and several accessory components, as depicted in Figure 2A. This design satisfied the static Taguchi's analysis requirements and was found suitable for the dynamic one, as in [12]. Its first (left) compartment was the main water tank manufactured from stainless steel, and the second (right) one was a control box with a digital step motor-driven water pump for water flow control. The top device in Figure 2A was an electrical protractor with an adjustable clamp to turn sonography probes by the required inclination angle. The above phantom simulated the blood flow in the iliac artery. This study adopted a 90 cm/s flow rate controlled by a noninvasive flow rate meter. The latter operated as the central electronic device, monitoring the water flow in the silicon pipe for scanning. Figure 2B depicts the water tank inner space ($L \times W \times H = 244 \times 194 \times 202$ mm$^3$), containing two silicon pipes with inner diameters of 19 and 10 mm and the wall thickness of 1 mm. Each pipe could be switched alternately by the external valve on the water tank bottom, marked in red in Figure 2A. As seen in Figure 2C, a 2 mm-thick silicon film shell of 175 mm in diameter was also used to simulate human skin: it was pressed tightly to the water surface. The water depth (preset at 45 mm) could be adjusted by the additional drain valve on the water tank bottom, marked in blue in Figure 2A. Since the pipe center was 20 mm from the bottom, the pipe top distance from the silicon film was 19 mm ($45 - 20 - (10 + 1) = 14$ mm) [12]. Figure 2D,E illustrate the adjustable clamp used to fix the required probe angle for scanning and an artificial stenosis analog model (silicon pipes of 50 mm in length and 8 or 4 mm inner diameter). These pipes simulated the stenosis by blocks with a tunnel cross-sectional area of 50.2 or 12.6 mm$^2$, respectively. The main advantage of the adopted phantom over other options was independent adjustability of each factor. Thus, flow rate, probe angle, and water depth were adjusted to the preset ultrasound scan protocol in order to arrange the $L_{18}$ orthogonal array for dynamic Taguchi's analysis. The particular settings are tabulated in Tables 1 and 2.

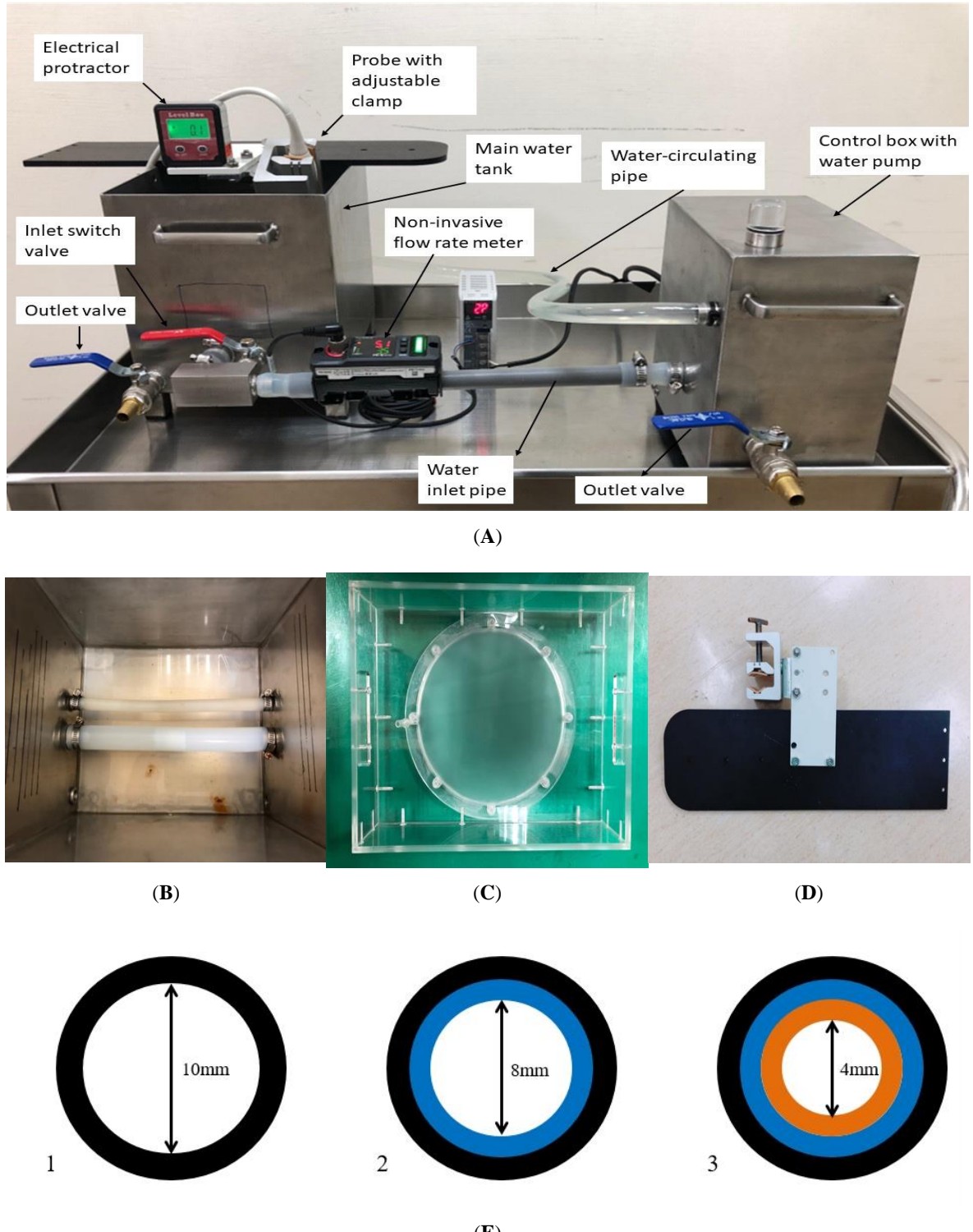

**Figure 2.** (**A**). The left part is the main water tank. The right part is the control box, whereas the central electronic device is a noninvasive flow rate meter. (**B**) The inner part of the water tank (L × W × H; 244 × 194 × 202 mm³). It includes two silicon pipes with inner diameters of 19 and 10 mm, and the 10 mm pipe is the target in this study; (**C**) 2 mm-thick silicon film 175 mm in diameter, simulating human skin, which could be pressed tightly over the water surface; (**D**) a close-up view of the adjustable clamp, which provides the required probe for scanning; and (**E**) the illustration of artificial stenosis made by silicon pipes similar to the pipe adopted in this work.

### 2.5. Quality Characteristics in Dynamic Taguchi Analysis

Unlike the static preset in Taguchi's optimization, the dynamic one is reputed for its strong flexibility in satisfying various boundary conditions in a clinical environment. The unique definition of sensitivity ($\beta$) allows one to correlate the expectation value, i.e., FOM versus the main variable. The latter may also involve various sizes of artifacts (i.e., the diameter of the stenosis of 10, 8, and 4 mm) as defined in this study or radioactive doses and phantom weights [14,20,21]. Nevertheless, the $\beta$ parameter is preset via the "higher-the-better" criterion to intensify good-quality characteristics in the conventional robust designation. In contrast, $\beta$ was redefined as "lower-the-better" to suppress the sensitivity of artifact sizes in this study and reduced to

$$\beta = \frac{sum\ of\ differences\ in\ group}{size\ of\ stenosis\ diameter} = \frac{Y}{X} = \frac{\sum_{j=1}^{m} \sum_{i=1}^{n} x_{ij} y_j}{\sum_{j=1}^{m} \sum_{i=1}^{n} x_{ij}^2} \quad (1)$$

$$stdev = \sqrt{\frac{\sum_{j=1}^{m} \sum_{i-1}^{n} \left(y_j - \beta x_{ij}\right)^2}{(n \times m) - 1}} \quad (2)$$

where $\beta$ is the sensitivity, $r^2$ is the correlation coefficient of the linear regression curve (sum of differences in groups vs. size of stenosis diameter), and $Y$ is the sum of differences between the practical survey and theoretical preset versus the specific stenosis diameter (i.e., 10, 8, or 4 mm, respectively, in this study) in 18 groups. Here, *stdev* is the standard deviation of the repeated calculations in obtaining the difference, averaged over $n$ (reviewers $\times$ repeated times) measurements for all $m$ diameters. In this study, $n = 9$ ($3 \times 3$) and $m = 3$.

### 2.6. Quantifying the Ultrasound Scanned Images

The artificial stenosis simulated by the 10 mm-diameter pipe was surveyed three times to ensure reproducibility. Thus, a total of 54 [$18 \times 3 = 54$] trials were scanned and recorded for one diameter, and we have three diameters (10, 8, and 4 mm) for further analysis in this study. The theoretical area of the artificial stenosis equaled 50.2 or 12.6 mm$^2$, respectively, and the original cross section without any artificial stenosis was 78.5 mm$^2$ ($\pi \times 5^2 = 78.5$), whereas the obtained data fluctuated given various combinations of assigned factors. Either systematic or random errors causing the fluctuation in the practical survey can be analyzed according to the FOM as proposed in this study. The ideal measured area of artificial stenosis should be 78.5, 50.3, or 12.6 mm$^2$ (ref. Figure 2E). Thus, the raw data were recorded to assess the difference between practical measurements and the theoretical value. In contrast, the integrated quality characteristic FOM was defined as revised from the original signal-to-noise ratio in Taguchi analysis;

$$FOM = r^2 / avg \cdot stdev^2 \cdot \beta^2 \quad (3)$$

The figure of merit (*FOM*) integrates the functional performance and quantifies the effect of the correlation coefficient ($r^2$), an average of raw data (*avg*), statistical standard deviation of the average (*stdev*), and sensitivity ($\beta$, cf. Equation (1)) altogether. A high FOM indicates a high $r^2$ or low *avg*, *stdev*, $\beta$ in the practical survey, which is preferable to satisfy the aim of this study.

## 3. Results

### 3.1. Raw Data Analysis

Table 3 summarizes the measured differences between the practical survey and theoretical preset of the cross section of artifacts (78.5, 50.2, or 12.6 mm$^2$ in this study), obtained standard deviation, β, correlation coefficient of the derived linear regression curve, FOM values obtained in three independent trials, and correlation coefficients $r^2$ derived via the MS EXCEL default linear regression function, altogether in 18 groups. As clearly illustrated, group 12 had the highest FOM among all.

**Table 3.** The evaluation results obtained from three independent measurements via Equations (1)–(3). The area difference is defined as the difference between the practical survey and theoretical ones (i.e., 78.5, 50.2, or 12.6 mm$^2$). Stdev values are the standard deviations obtained in each group from three trials (cf. Equation (2)).

| Group | 78.5 mm$^2$ | | | 50.2 mm$^2$ | | | 12.6 mm$^2$ | | | Avg. | Stdev | β | $r^2$ | FOM |
|---|---|---|---|---|---|---|---|---|---|---|---|---|---|---|
| | #1 | #2 | #3 | #1 | #2 | #3 | #1 | #2 | #3 | | | | | |
| 1 | 1.50 | 10.50 | 3.50 | 3.800 | 1.200 | 5.800 | 0.400 | 3.400 | 0.600 | 3.411 | 3.194 | 6.0 | 0.990 | **0.015** |
| 2 | 13.50 | 10.50 | 7.50 | 0.200 | 0.800 | 4.800 | 1.400 | 4.400 | 0.400 | 4.833 | 4.792 | 12.0 | 0.558 | **0.002** |
| 3 | 8.50 | 7.50 | 12.50 | 0.200 | 2.200 | 0.200 | 1.400 | 0.400 | 1.600 | 3.833 | 4.501 | 11.9 | 0.544 | **0.003** |
| 4 | 7.50 | 3.50 | 12.50 | 0.800 | 6.200 | 5.800 | 2.400 | 2.400 | 0.600 | 4.633 | 3.810 | 9.5 | 0.916 | **0.005** |
| 5 | 4.50 | 1.50 | 4.50 | 1.800 | 1.200 | 3.800 | 1.600 | 0.400 | 0.600 | 2.211 | 1.618 | 4.3 | 0.977 | **0.064** |
| 6 | 2.50 | 8.50 | 8.50 | 4.200 | 1.200 | 0.800 | 3.400 | 2.400 | 0.400 | 3.544 | 3.059 | 6.3 | 0.571 | **0.008** |
| 7 | 2.50 | 1.50 | 7.50 | 2.200 | 3.200 | 0.200 | 0.400 | 2.400 | 1.600 | 2.389 | 2.150 | 3.5 | 0.722 | **0.040** |
| 8 | 2.50 | 1.50 | 5.50 | 4.200 | 5.200 | 1.200 | 2.400 | 4.400 | 0.400 | 3.033 | 1.849 | 1.5 | 0.628 | **0.075** |
| 9 | 1.50 | 6.50 | 4.50 | 2.200 | 2.200 | 3.800 | 0.600 | 0.400 | 0.600 | 2.478 | 2.077 | 6.0 | 0.995 | **0.032** |
| 10 | 1.50 | 4.50 | 0.50 | 6.200 | 2.200 | 1.800 | 0.400 | 2.400 | 0.600 | 2.233 | 1.958 | 2.3 | 0.379 | **0.038** |
| 11 | 1.50 | 1.50 | 8.50 | 1.200 | 2.800 | 5.800 | 1.400 | 0.400 | 0.600 | 2.633 | 2.733 | 5.2 | 0.976 | **0.026** |
| 12 | 0.50 | 0.50 | 3.50 | 2.200 | 0.200 | 2.800 | 1.400 | 0.400 | 1.400 | 1.433 | 1.176 | 0.9 | 0.600 | **0.413** |
| 13 | 2.50 | 3.50 | 0.50 | 4.200 | 11.200 | 0.800 | 2.400 | 1.400 | 0.600 | 3.011 | 3.334 | 2.4 | 0.123 | **0.005** |
| 14 | 1.50 | 3.50 | 1.50 | 0.800 | 3.200 | 2.200 | 0.600 | 0.600 | 0.600 | 1.611 | 1.129 | 2.8 | 0.923 | **0.184** |
| 15 | 0.50 | 4.50 | 0.50 | 1.800 | 1.200 | 0.800 | 0.400 | 1.600 | 1.600 | 1.433 | 1.268 | 0.9 | 0.664 | **0.393** |
| 16 | 10.50 | 6.50 | 7.50 | 7.200 | 4.200 | 4.200 | 2.600 | 0.400 | 0.600 | 4.856 | 3.372 | 11.4 | 0.989 | **0.005** |
| 17 | 5.50 | 6.50 | 1.50 | 2.200 | 4.200 | 0.800 | 0.400 | 2.400 | 1.400 | 2.767 | 2.147 | 4.8 | 0.854 | **0.030** |
| 18 | 5.50 | 5.50 | 2.50 | 1.200 | 10.200 | 3.200 | 0.400 | 2.400 | 0.600 | 3.500 | 3.132 | 6.1 | 0.832 | **0.012** |

### 3.2. Inspecting the Domination of Factor

Rearranging and averaging avg, stdev, and FOM values for each group under three sizes of artifacts was performed for various factors from Table 1. Thus, averages for groups 1–9 and 10–18 are contributions by the angle of probe (degree). Averages of derivation for (1, 2, 3, 10, 11, and 12), (4, 5, 6, 13, 14, and 15), and (7, 8, 9, 16, 17, and 18) groups revealed the contributions to the difference of the various levels of factor B (signal gain). Additionally, the signal gain was set to 50 (B1), 55 (B2), and 60 (B3) herein (cf. Table 2). The rearranged values against the scanned image factors of the ultrasound facility are plotted in Figure 3. As seen in Table 3, group 12 exhibited the highest FOM value (FOM = 0.413) among all groups, featuring avg = 1.433 and stdev = 1.176. Fluctuations associated with changes in factors A (angle of probe), B (signal gain), C (resolution vs. speed), E (XRES), and G (time gain compensation) exceeded those of the remaining factors. Therefore, the ultrasound scan imaging system performance was influenced by factors A, B, C, E, or G in this study.

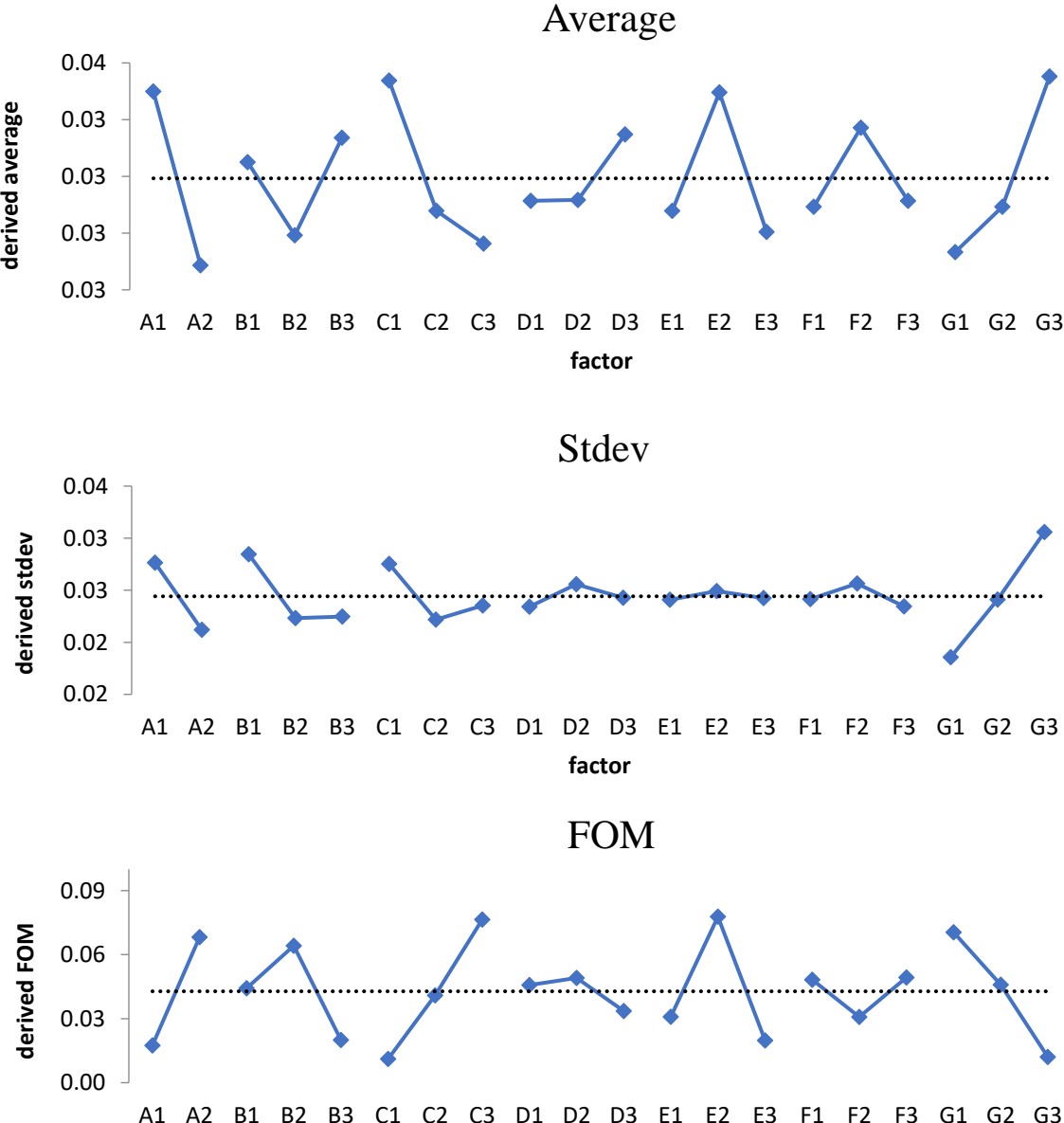

**Figure 3.** The respective three fishbone plots of the ultrasound scan protocol. Factors A (angle of probe), B (signal gain), C (resolution vs. speed), E (XRES), and G (time gain compensation) have the dominant contribution among all factors (cf. Table 3).

## 4. Discussion

### 4.1. Taguchi's Analysis Verification

Taguchi's analysis provides robust designation and is famous for its capability in fulfilling practical needs. Thus, it is essential to have further verification or testification after the preliminary survey from the obtained fishbone plots. Without an additional step of follow-up verification or testification, the optimal suggestion might not pass the clinical application [16,21]. Accordingly, four kinds of factor combinations were organized to verify the optimal suggestion, namely A—the original G12, the highest FOM among the 18 groups, denoted as A2B1C3D2E2F1G1, where the capital letter implies the factor and the digit implies the level of that specific factor (ref. Tables 1 and 2); B—the conventional combination that previous users suggested, denoted as A2B1C2D2E2F2G2; C—the combination of the highest FOM of each factor, denoted as A2B2C3D2E2F1G1; D or E—revised from

group C considering the cross-interaction among factors I, denoted as A2B2C2D1E2F1G1; and II, denoted as A1B2C2D2E2F1G1. The practical scanning and derived outcomes are listed in Table 4. As clearly illustrated, the original group 12 (i.e., group A in this comparison) still holds the maximal FOM; thus, the combination of A2B1C3D2E2F1G1 proved to be the optimal suggestion to scan the carotid artery stenosis of patients in this study. Taguchi analysis continually compiles every single quality characteristic into a compromised performance; thus, intensifying only a single feature is not the solution from the viewpoint of robust designation, whereas FOM, as proposed in this study, can provide an integrated consideration in clinical diagnosis to give a superior imaging quality of ultrasound scan. Note that any new suggested combination of ultrasound scan facilities has superior image quality than the conventional one, indicating that an inappropriate protocol may mislead the judgment in clinical diagnosis.

**Table 4.** Four groups of factor combinations of ultrasound facility to verify the optimal suggestion according to different scenarios: A. original group 12, the highest FOM among 18 groups; B. conventional suggestion; C. the combination of highest FOM of each factor; D. considering the cross-interaction among factors I; and E. II.

|  | Group A (Original G12) | Group B (Conventional) | Group C (Highest FOM of Each Factor) | Group D (Considering Cross-Interaction I) | Group E (Considering Cross-Interaction II) |
|---|---|---|---|---|---|
| **Factor:** | | | | | |
| (A) angle of probe (degree) | 0 | 0 | 0 | 0 | 30 |
| (B) signal gain (dB/cm) | 50 | 50 | 55 | 55 | 55 |
| (C) resolution vs. speed | S1 | RS | S1 | RS | RS |
| (D) dynamic range (dB) | 50 | 50 | 50 | 40 | 50 |
| (E) XRES (scenario) | 2 | 2 | 2 | 2 | 2 |
| (F) zoom | 0.8 | 1.0 | 0.8 | 0.8 | 0.8 |
| (G) time gain compensation | −1 | 0 | −1 | −1 | −1 |
| **Area:** | | | | | |
| Difference in 78.5 mm$^2$ | 1.50 | 8.5 | 3.50 | 1.50 | 3.50 |
| Difference in 50.2 mm$^2$ | 1.73 | 2.8 | 1.20 | 0.20 | 1.20 |
| Difference in 12.6 mm$^2$ | 1.07 | 0.6 | 6.40 | 4.40 | 5.40 |
| Avg. | 1.43 | 3.97 | 3.70 | 2.03 | 3.37 |
| Stdev | 1.18 | 4.08 | 2.61 | 2.15 | 2.10 |
| $\beta$ | 0.86 | 12.07 | 6.00 | 5.64 | 4.21 |
| $r^2$ | 0.60 | 0.82 | 0.50 | 0.64 | 0.38 |
| **FOM** | **0.413** | **0.004** | **0.012** | **0.031** | **0.013** |
| **Diameter:** | | | | | |
| Difference in 10 mm | 0.50 | 0.70 | 0.003 | 0.30 | 0.30 |
| Difference in 8 mm$^2$ | 0.20 | 0.02 | 0.002 | 0.00 | 0.38 |
| Difference in 4 mm$^2$ | 0.11 | 0.47 | 0.001 | 0.15 | 0.00 |
| Avg. | 0.27 | 0.40 | 0.002 | 0.15 | 0.23 |
| Stdev | 0.33 | 0.35 | 0.001 | 0.15 | 0.20 |
| $\beta$ | 0.662 | 0.168 | 0.003 | 0.16 | 0.56 |
| $r^2$ | 0.844 | 0.022 | 0.964 | 0.108 | 0.740 |
| **FOM** | **14.24** | **0.95** | **1.61E8** | **29.81** | **29.00** |

## 4.2. The FOM Focused on the Difference in Area or Diameter

It is an alternative suggestion for ultrasound diagnosis of carotid artery stenosis in patients to scan the stenosis diameter. In doing so, the same combination of factors as listed in Table 1 was adopted to measure the difference in the practical survey and theoretical preset in this study (i.e., 10, 8, and 4 mm). As clearly implied in the bottom part of Table 3, group C has an extremely high index of FOM, indicating outstanding performance in analyzing the diameter of stenosis. In contrast, group A (i.e., original group 12 in the $L_{18}$ orthogonal array, cf. Table 2) does not perform better than the others. This is because the presumption of this study was preset to optimize the imaging quality of the area, not diameter; therefore, the demanded expectation cannot be arbitrarily switched in the domain of robust designation methodology [22,23]. This is crucial in applying Taguchi analysis to a single expectation demand in reality. Moreover, the conventional setting of ultrasound scan facilities still cannot have the convincing capability to recognize the solid diameter.

Low FOMs of the conventional setting show an inappropriate setting in identifying either area or diameter in a routine examination.

### 4.3. The FOM Advantages over the S/N Ratio

The quality characteristic as defined in this study is FOM (cf. Equation (3)), which integrates not only the difference between the practical survey and theoretical preset but also the average, stdev, β, and correlation coefficient altogether. The FOM shows its excellent capability in compiling the individual imaging-quality characteristics and reveals a good suggestion with many quantified values, although the Taguchi suggested signal-to-noise ratio in dynamic analysis still has its advantage in practical consideration. Define the signal-to-noise as Equation (4) to revise the derived data in this study;

$$\frac{S}{N}(\eta_i) = -10 \log\left[(stdev/avg)^2\right] \qquad (4)$$

where stdev is the standard deviation as defined in Equation (2), and avg is the average of individual differences between practical measurements and theoretical preset in this study. As clearly depicted in Figure 4, the dominant factors become factor B (signal gain), C (resolution vs. speed), D (dynamic range), E (XRES), and G (time gain compensation). The trend of each factor has significantly different behaviors, as implied in Figure 3. Factor A (angle of probe) becomes minor, whereas D (dynamic range) becomes dominant. Note that S/N focuses only on the stdev and average of the difference and ignores other contributions from β (sensitivity) or $r^2$ (correlation coefficient) (cf. Equation (3)) in reality. Accordingly, the newly proposed FOM might be an alternative option in correlate research since it compiles more imaging quality characteristics in reality.

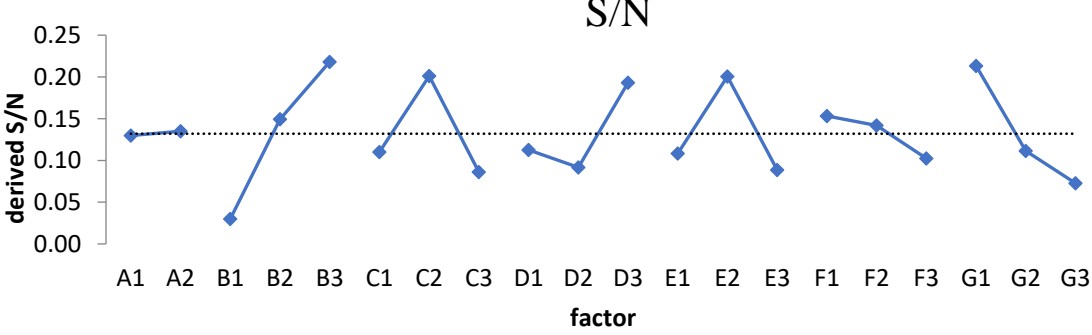

**Figure 4.** The fishbone plot of the S/N (cf. Equation (4)). The dominant factors become B (signal game), C (resolution vs. speed), D (dynamic range), E (XRES), and G (time gain compensation).

### 4.4. Clinical Testification

An ideal solution to solidify the Taguchi suggestion of any specific topic is either to verify or to testify in reality [13,24]. Therefore, three well-trained radiologists majoring in collecting ultrasound scan images were invited to be the reviewers of 32 pairs of scanned images taken from five volunteers on either the left or right carotid artery. Each pair of images had two corresponding images scanned from conventional or optimal settings, which were randomly submitted to the reviewers and graded as "fail" or "pass" according to their clearness and sharpness. Accordingly, the surveyed results are arranged in Table 5, and 5 out of 32 pairs of ultrasound scanned images are depicted in Figure 5. As clearly denoted in Table 5, a total of 192 images were graded (32 × 3 × 2 = 192), and the conventional combination of ultrasound preset gained 30 passes versus 66 passes for the optimal combination. Specifically, when each pair of scan slices was compared, 17 pairs of both conventional and optimal slices, 13 pairs of only optimal slices, and 2 pairs of only conventional slices passed the clinical criteria. Figure 5 also reveals the corresponding scan

slices in this study. Pair A–D shows only one optimal pass, whereas pair E shows only one conventional pass that passed the clinical criteria.

**Table 5.** Three well-trained radiologists graded the statistical data of 32 scanned images from 5 volunteers. Thus, a total of 192 ($32 \times 3 \times 2 = 192$) images were graded by three radiologists.

| Pair Numbers | Conventional | Optimal |
| --- | --- | --- |
| 17 | Pass | Pass |
| 13 | Fail | Pass |
| 2 | Pass | Fail |
| Total: 32 | 30 Pass | 66 Pass |

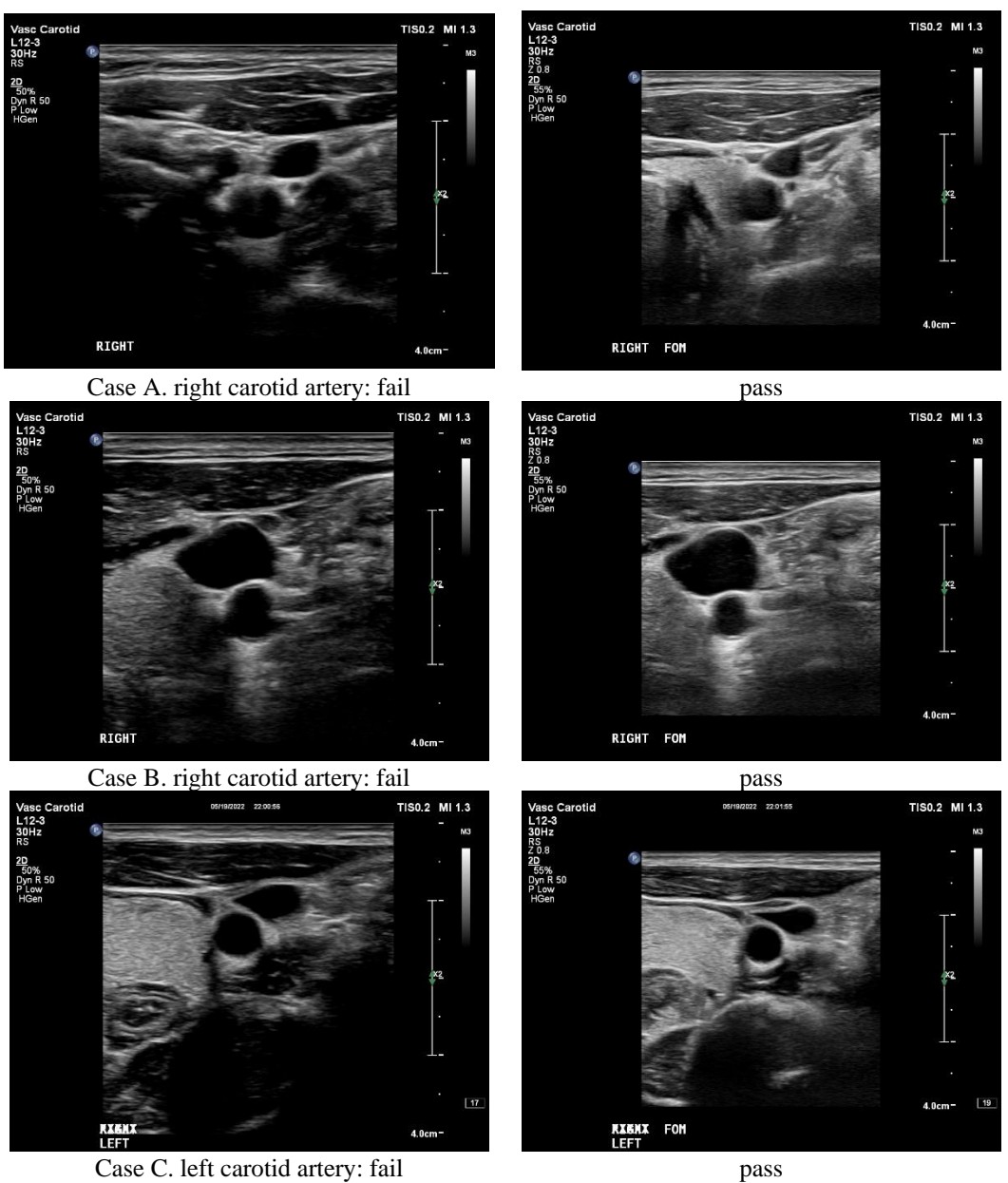

Case A. right carotid artery: fail        pass

Case B. right carotid artery: fail        pass

Case C. left carotid artery: fail        pass

**Figure 5.** *Cont.*

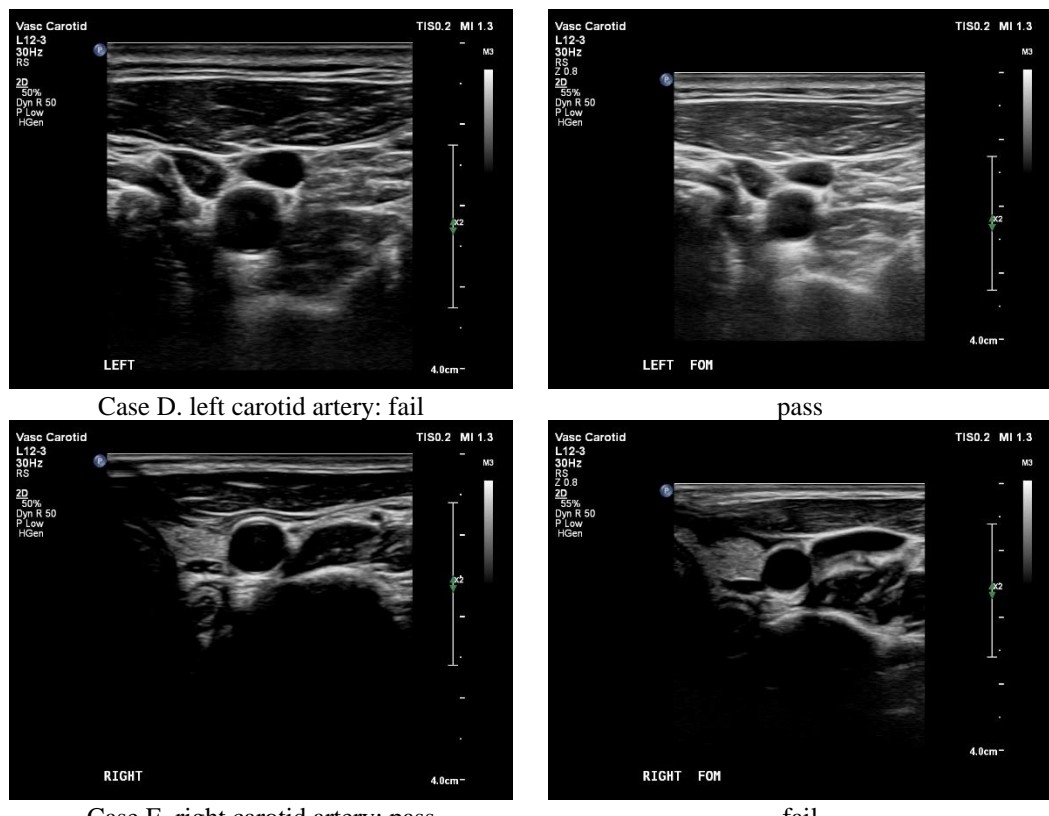

Case D. left carotid artery: fail          pass

Case E. right carotid artery: pass          fail

**Figure 5.** The ultrasound scanned images in this study. Pair (**A**–**D**) shows only one optimal pass, whereas pair (**E**) shows only one conventional pass that passed the clinical criteria.

## 5. Conclusions

This study optimized the ultrasound image of carotid artery stenosis using dynamic Taguchi analysis and an indigenous water phantom. In doing so, an innovative quantitative FOM index was proposed to evaluate the integrated performance of ultrasound scanned images. A silicon pipe customized the artifacts of stenosis in the carotid artery, and the artifacts could be quantified by either the diameter or area of the cross section from the scanned images. Accordingly, eighteen combinations of seven factors in operating the ultrasound facility were organized according to the Taguchi $L_{18}$ orthogonal array, and the difference in the derived area between the practical survey and theoretical preset of stenosis in each group was combined with stdev, β, and $r^2$ to derive the FOM. Furthermore, the same data were reorganized to derive the fishbone plot of every specific factor according to the Taguchi suggestion. The correlated discussions focused on the innovated FOM and the original S/N ratio in the Taguchi analysis. The follow-up study also mentioned the alternative option of adopting the difference in the diameter as the aim of optimization.

**Author Contributions:** Formal analysis, K.-Y.H.; Investigation, C.-S.L. and W.-M.L.; Methodology, S.-H.H., Y.-T.C. and B.-R.P.; Writing—original draft, L.-K.P.; Writing—review & editing, L.-F.P. All authors have read and agreed to the published version of the manuscript.

**Funding:** This research received no external funding.

**Institutional Review Board Statement:** The TAFGH Institutional Review Board committee approved the study with credential No. TSGHIRB 2-105-05-089, and the requirement for informed consent was waived.

**Informed Consent Statement:** Informed consent was obtained from all subjects involved in the study.

**Conflicts of Interest:** The authors declare no conflict of interest.

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
