# Peer review of "Optimizing the Ultrasound Image Quality of Carotid Artery Stenosis Patients via Taguchi’s Dynamic Analysis and an Indigenous Water Phantom"

_applsci, doi:10.3390/app12199751_

Round 1

Reviewer 1 Report

In this study, the author proposed a new method, dynamic analysis, to optimize ultrasound image. Through in vivo experiment, “more pass” were obtained using proposed method in contrast of conventional method. This is kinds of interesting finding. However, the manuscript wasn’t organized well. For example, how to perform the dynamic analysis wasn’t explained clearly. It is kinds of hard to follow the key idea. A flow chart was recommended for better understand and recognized the manuscript.

Comments:

1.     Please provide full name of XRES.

2.     In the introduction section, the author reviews many current studies and claimed that “ optimizing image quality of grayscale B-mode carotid ultrasound scans of carotid artery stenosis patients remains a challenging problem”. Please provide further detail on the challenging and limitations of reported studies.

3.     Please provide x lable and y lable for Figures 2 and 3.

4.     Using this method, the image quality would be improved. A comparison will be helpful. For example, two US images from different methods.

5.     In Figures 2 and 3, for A-G groups, three measurements were preformed except A, is there any specific reason?

6.     Three well-trained radiologists performed review. What is the general standard for pass or fail? Please provide further discussion. Which attribute of US image obtained from proposed method contribute to more probability of pass?

7.     The key innovation of manuscript is dynamic Taguchi analysis, it will be helpful if the author provides a flow chart to demonstrate the detailed image processing protocol.

8.     It is recommended to use figure to replace table.

9.     For US images in Figure 4, please label the interest of area.

Reviewer 2 Report

The authors present the application of Taguchis analysis to improve medical diagnostics of carotid artery stenosis. The study was conducted in close relationship to physicians in the field. The used statistical approach seems to be very valuable, as the interpretation of ultrasound data is strongly influenced by the experience of the clinician and anatomical features, making it hard to provide generic recommendations or settings. Some open questions can still be addressed in order to improve the genericity of the findings:

- Other works are only mentioned in Sec. 1, but this work is not really embedded into the state of the art. Furthermore, studies that utilize Taguchis analysis in a similar matter are not mentioned.
- Please provide a deeper explanation of the used method in Sec. 2.2. It is also not clear why these exact orthogonal arrays were used.
- The study is very bound to the specific scanner used (Philips Affiniti). A more thorough specification of the used setup should be given (frequency, bandwidth, imaging method and rate) in order to compare these results. Furthermore, the usage of the results of this study with other setups should be discussed.
- The setup was optimized with a static phantom. The influence of individual anatomical features should be discussed more thoroughly, as this can limit the scope of the proposed method.
- Give some more explanations in 2.6 on how exactly the stenosis was quantified, i.e. how the study was designed.
- Tab. 2 (B) is the signal amplificiation gain static (therefore in dB) or depending on the depth (dB/cm)? For the latter, how is this different to time gain compensation?
- Tab. 3: What is Eq. 6?
